DATA RELEASE

# The assembly and annotation of two teinturier grapevine varieties, Dakapo and Rubired

Eleanore J. Ritter[1], Noé Cochetel[2], Andrea Minio[2], Peter Cousins[3], Dario Cantu[2,4,*] and Chad Niederhuth[1,*,†]

1 Department of Plant Biology, Michigan State University, East Lansing, MI 48824, USA
2 Department of Viticulture and Enology, University of California Davis, Davis, CA 95616, USA
3 E. & J. Gallo Winery, Modesto, CA 95354, USA
4 Genome Center, University of California Davis, Davis, CA 95616, USA

## ABSTRACT

Teinturier grapevines, known for their pigmented flesh berries due to anthocyanin production, are valuable for enhancing the pigmentation of wine, for potential health benefits, and for investigating anthocyanin production in plants. Here, we assembled and annotated the Dakapo and Rubired genomes, two teinturier varieties. For Dakapo, we combined Nanopore sequencing, Illumina sequencing, and scaffolding to the existing grapevine assembly to generate a final assembly of 508.5 Mbp. Combining *de novo* annotation and lifting over annotations from the existing grapevine reference produced annotation 36,940 gene annotations for Dakapo. For Rubired, PacBio HiFi reads were assembled, scaffolded, and phased to generate a diploid assembly with two haplotypes 474.7–476.0 Mbp long. *De novo* annotation of the diploid Rubired genome yielded annotations for 56,681 genes. Both genomes are highly contiguous and complete. The Dakapo and Rubired genome assemblies provide genetic resources for investigations into berry flesh pigmentation and other traits of interest in grapevine.

**Subjects** Genetics and Genomics, Plant Genetics, Bioinformatics

Submitted: 17 June 2024

* Corresponding authors. E-mail: dacantu@ucdavis.edu; niederhuth.kox6g@passmail.net

† Current Address: Corteva, Inc. Indianapolis, IN 46268, USA.

Preprint submitted at https://doi.org/10.1101/2024.05.03.592245

## CONTEXT

Domesticated grapevine (*Vitis vinifera*) is the fifth most produced fruit globally [1], with 80.1 million tonnes produced in 2022 alone [2]. Various grapevine varieties have been bred since its estimated domestication ~11–15,000 years ago [3, 4], for both consumption as table grapes and winemaking purposes. This has resulted in the selection of numerous diverse phenotypes with significant variation in traits, including berry color and aromatic compounds, as well as more utilitarian traits like yield or biotic and abiotic stress resistance. The berry color is of particular importance in wine grapes due to how it influences wine color and quality. Generally, both consumers and experts prefer red wines with darker colorations [5, 6], making strong pigmentation in berries advantageous for wine producers. Pigmentation typically occurs only in the skin of ripened grapevine berries, with most grapevine varieties having white-colored flesh. The pigmentation within the berry skin is due to the production of anthocyanins, which are colored flavonoids that also act as antioxidants [7]. As anthocyanins significantly influence both the quality of wines and their health benefits, the genetic and molecular pathways involved in anthocyanin produced in berry skin have been of high interest and well-characterized [8].

**Figure 1.** The ancestry of Dakapo and Rubired, with berry skin and flesh color shown. Dakapo and Rubired are thought to have been bred from Teinturier du Cher clones with differing copy numbers of a 408 bp repeat within the promoter of *VvMybA1*, which is noted in the figure.

Teinturier (also known as "dyer") varieties produce berries with pigmented skin and flesh, as well as pigmented leaves. They are highly favorable for use in red wine blends, as they provide a deeper color. They also remain valuable resources for understanding the production of anthocyanins outside of berry skin. Dakapo and Rubired are two teinturier varieties that are widely grown, and Rubired was the 8th most crushed grapevine variety in California in 2022 [9]. Both varieties are descendants of Teinturier du Cher, a teinturier grape variety used in the 19th century to breed most teinturier varieties existing today, but of distinct generations. Dakapo was initially bred through a cross between Deckrot and Blauer Portugieser, with Deckrot being a direct descendant of Teinturier du Cher. Rubired is a hybrid grapevine variety bred through a cross between Tinto Cão and Alicante Ganzin, with Alicante Gazin being a fourth-generation descendent of Teinturier du Cher. While both Dakapo and Rubired are descendants of Teinturier du Cher, their ancestors were likely distinct clones of Teinturier du Cher based on previous genetic work in teinturier grapes [10] (Figure 1).

Teinturier varieties have substantially higher anthocyanin content in their berries than non-teinturier varieties due to the accumulation of anthocyanins within their berry flesh,

which results in berry flesh pigmentation. Previous work showed that the juice produced with Dakapo berries had 39–91 times more anthocyanin content than commercial red grape juice [11]. Teinturier varieties themselves vary in anthocyanin content and the profiles of anthocyanins present within the berry flesh [10, 12]. Previous studies have made progress in investigating the genetic basis of berry flesh pigmentation and variation in overall anthocyanin production within teinturier grapes. A previous study [10] demonstrated that increased copies of a 408 bp repeat in the promoter of the gene *VvMybA1*, known as the grapevine color enhancer (GCE), is directly linked to increased anthocyanin production in teinturier berries. *VvMybA1* plays a significant role in regulating anthocyanin production alongside *VvMybA2*, and many berry color mutants are the result of mutations impacting *VvMybA1* [13–16]. A single copy of this 408 bp repeat is present upstream of *VvMybA1* alleles for red- and white-skinned grapes with unpigmented flesh as well [10]; however, the allele responsible for white berry skin color also contains a *Gret1* retrotransposon upstream of coding sequences [13]. In the allele responsible for white berry skin color, this *Gret1* retrotransposon is present between *VvMybA1* and the GCE, and is thought to block the expression of *VvMybA1*, causing a loss of pigmentation in berry skin [13]. Previous work [10] demonstrated that the copy number of this 408 bp sequence varied between varieties and that varieties derived from Teinturier du Cher had either two, three, or five copies of this 408 bp sequence within the promoter region, with Rubired and Dakapo having alleles with two and three copies of the repeat, respectively. More than one copy of this repeat in tandem was associated with berry flesh pigmentation in teinturier grapes. Increased copies of these repeats were correlated with increased expression of *VvMybA1* and increased anthocyanin content within berry skin, berry flesh, and leaves [10]. Additional past work demonstrated that the alleles enabling berry flesh pigmentation in teinturier varieties appear to be dominant [17]. While past work greatly illuminated the genetic basis of increased anthocyanin production in Teinturier du Cher descendants, it is still unclear why different teinturier grapes have distinct anthocyanin profiles in berry flesh, regardless of the number of copies of the 408 bp sequence they have [12]. While overall anthocyanin content does correlate with the number of 408 bp repeats upstream of *VvMybA1* [10], large differences in concentrations of specific anthocyanins exist between teinturier varieties with the same number of copies of the 408 bp repeat [12]. For example, the concentration of a specific type of anthocyanin, Cyanidin-3-*O*-glucoside, can vary from 1.0 to 21.0 mg/L when comparing the anthocyanin content within the flesh of teinturier berries from different varieties that all contain two copies of the 408 bp repeat [12]. The assembly and annotation of the genome of the Yan73 teinturier grapevine variety, which contains three copies of this 408 bp repeat, were recently generated and provided additional insight into the regulation of anthocyanin accumulation in Yan73 berry flesh [18]. However, the lack of additional genomic resources for teinturier grapes has inhibited further investigations into differences between teinturier varieties, and the genetic basis for these large differences in anthocyanin composition remains unclear.

Here, we sequenced, assembled, and annotated the Dakapo and Rubired genomes to provide additional resources for understanding teinturier varieties and to further enable their use in breeding programs. These genomes will greatly facilitate future work into understanding the regulation of anthocyanins within berry flesh. Beyond anthocyanin production, Dakapo and Rubired have also been utilized to research other traits in grapevine. A QTL mapping population of Dakapo × Cabernet Sauvignon) has been

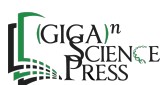

established and utilized to investigate *Botrytis* bunch rot in grapevine [19]. Additionally, Rubired is notable for being highly resistant to *Xylella fastidiosa*, which causes Pierce's disease in grapevine [20–22]. As a result, we believe these high-quality reference genome assemblies and annotations will be a useful resource for the grapevine and plant science communities.

## METHODS

### Plant material

*Vitis vinifera* plants of the Dakapo variety were planted in Madera, California, USA in 2011. Young leaf tissue samples for Oxford Nanopore Technologies (ONT) long-read sequencing were collected in July 2021. The samples were frozen and shipped on dry ice overnight. Plant material used in this study was also utilized in a previous study [23] as the "Dakapo WT" samples. For Rubired tissue, young leaves were collected from the accession Rubired Foundation Plant Services (FPS) clone 02 maintained by the Foundation Plant Services at the University of California, Davis.

### DNA extraction and sequencing of Dakapo tissue

High molecular weight DNA was extracted, and a sequencing library was prepared for ONT sequencing by the Genomics Core at Michigan State University, as previously described [23], using the Oxford Nanopore Technologies Ligation Sequencing Kit (SQK-LSK109). The library was sequenced on a PromethION FLO-PRO002 flow cell (R9.4.1; Oxford Nanopore Technologies) on a PromethION24 (Oxford Nanopore Technologies) running MinKNOW Release 21.11.7 (Oxford Nanopore Technologies) [24], resulting in 61.9 Gbps of sequence (~123.8× coverage) with a read length N50 of 13.2 kbp. Base calling and demultiplexing were performed using Guppy v5.1.13 (RRID:SCR_023196, Oxford Nanopore Technologies) with the High Accuracy base calling model. An additional 9.5 Gbps (~18.9× coverage) of ONT sequencing data and 25.6 Gbps (~51.3× coverage) of Illumina paired-end sequencing data previously published from the "Dakapo WT" sample described previously [23] were also utilized for this study (available on the NCBI Sequence Read Archive under BioProject PRJNA1020818; complete sequencing statistics in Supplementary Table S1 on GigaDB [25]). The additional ONT sequencing data used from previously published work [23] was generated using the same tissue and methods described here.

### Dakapo genome assembly

Raw ONT sequencing data from this study and previous work with the same plant material [23] were combined. Adapters were trimmed using Porechop v0.2.4 [26] with the following settings: *--min_trim_size 5, --extra_end_trim 2, --end_threshold 80, --middle_threshold 90, --extra_middle_trim_good_side 2, --extra_middle_trim_bad_side 50,* and *--min_split_read_size 300*. Reads mapping to the lambda phage genome were removed using NanoLyse v1.2.0 (RRID:SCR_024125) [27]. NanoFilt (RRID:SCR_016966) v2.8.0, with the flags *-q 0* and *-l 300,* was used to remove low-quality reads and reads shorter than 300 base pairs (bp) [27]. The quality of the reads was analyzed using FastQC v0.11.9 (RRID:SCR_014583) [28], NanoStat v1.6.0 [27], and NanoPlot v1.38.0 (RRID:SCR_024128) [29]. ONT reads were then assembled using Flye v2.8.3-b1695 (RRID:SCR_017016) [30] for two iterations. One round of polishing was performed on the assembly using the ONT reads with Racon v1.4.20 (RRID:SCR_017642) [31] and the following settings: *--include-unpolished,*



*-m 8, -x -6, -g -8*, and *-w 500*. The assembly was then scaffolded to the 12X.v2 grapevine genome assembly [32] using RagTag v2.0.1 "scaffold" [33] with the following settings: *-f 1000, -d 100000, -i 0.2, -a 0.0, -s 0.0, -r, -g 100*, and *-m 100000*.

Paired-end Illumina reads were used for the final polishing of the scaffolded assembly. These were first trimmed using Trimmomatic v0.39 (RRID:SCR_011848) [34] to remove adapters and low-quality sequences, with the following settings: *-phred33, ILLUMINACLIP:TruSeq3-PE-2.fa:2:30:10:4:TRUE, LEADING:3, TRAILING:3, SLIDINGWINDOW:4:15*, and *MINLEN:30*. The reads were then mapped to the draft genome assembly using BWA-MEM v0.7.17-r1188 (RRID:SCR_010910) [35] with the *-M* flag. PCR duplicate reads were removed using Picard MarkDuplicates v2.15.0 [36] with the *−−REMOVE_DUPLICATES TRUE* flag. The mapped reads with marked duplicates were then used to polish the draft assembly using Pilon v1.24 (RRID:SCR_014731) [37] with the *−−fix all* flag used to correct all errors identified and the *−−diploid* flag. Two iterations of Pilon polishing were performed.

Following polishing, haplotigs were removed using Purge Haplotigs v1.1.2 [38]. To do so, all prepped ONT reads were mapped to the Dakapo draft assembly using minimap2 v2.23-r1111 (RRID:SCR_018550) [39] with the flags *-ax map-ont* and *-L*, and *purge_haplotigs hist* was then run with default settings to generate a read-depth histogram of these mapped reads. Based on the histogram generated, *purge_haplotigs cov* was run with the previous output file and the following flags: *-low 15, -mid 88*, and *-high 195*. Finally, *purge_haplotigs purge* was run using the previous output file to purge haplotigs from the Dakapo draft assembly [38].

Before finalizing the Dakapo genome assembly, chr00 was split apart manually at gaps since it is an artificial chromosome of unmapped contigs from the 12X.v2 grapevine genome assembly [32] used for scaffolding. The assembly was also searched for microbial contamination using the gather-by-contig.py script adapted from [40], which utilizes sourmash (RRID:SCR_024347) and its pre-built database "GTDB R06-RS202 genomic representatives" [41]. No contamination was found from this process. The chromosome names were maintained from the scaffolding to the 12X.v2 grapevine genome assembly [32]. All other contigs were sorted and renamed in order of length, including the contigs split apart from chr00, using the custom script sort_rename_fasta.sh.

To assess the quality of the Dakapo genome assembly, we used BUSCO v5.2.2 (RRID:SCR_015008) [42] to check the completeness of the assembly when compared to the eudicots_odb10 dataset at each step of genome assembly and polishing. Assembly statistics were calculated using assembly-stats v1.0.1 (RRID:SCR_023963) [43]. Finally, the quality of repetitive sequences and intergenic space was also assessed by calculating the long terminal repeats (LTR) Assembly Index (LAI) for the Dakapo assembly using LTRs annotated by the Extensive *de-novo* transposable element (TE) Annotator (EDTA) (TE annotation methods are reported below).

## Dakapo genome annotation

TEs and repeats in the Dakapo genome assembly were annotated using EDTA v1.9.4 (RRID:SCR_022063) [44] with the following flags: *−−species others, −−step all, −−overwrite 1, −−sensitive 1, −−anno 1, −−evaluate 0*, and *−−force 1*.

MAKER (RRID:SCR_005309) was used for *de novo* annotation of genes in the Dakapo genome. Before running MAKER, RNA-seq reads from diverse tissues in grapevine

(including leaves, seeds, fruits, roots, and various floral components) and protein sequences from related species were used to provide the initial support for gene models. To do so, RNA-seq samples from previous studies [45–50] were downloaded from the NCBI Sequence Read Archive (SRA) using fasterq-dump v2.10.7 from the sra-toolkit (RRID:SCR_024350) [51] (Supplementary Table S2 on GigaDB [25] shows the SRA IDs of the specific files used). Trimmomatic v0.39 [34] was used to trim adapters from Illumina RNA-seq reads with the flags: *−−phred33* and *ILLUMINACLIP:TruSeq3-PE-2.fa:2:30:10*. These were then mapped to the Dakapo genome assembly using HISAT2 v2.2.1 (RRID:SCR_015530) [52] with the *−−phred33* flag. PacBio RNA-seq reads were mapped to the Dakapo genome assembly using minimap2 v2.23-r1111 [39] with the flags *-ax splice:hq* and *-uf*. Transcripts from these mapped RNA-seq reads were assembled using StringTie v2.2.1 (RRID:SCR_016323) [53] with the following flags: *-c 1, -f 0.01, -m 200, -a 10, -j 1, -M 1, -s 4.75* (for mapped Illumina reads) or *1.5* (for mapped PacBio reads), and *-g 50* (for mapped Illumina reads) or *0* (for mapped PacBio reads). The output files for all RNA-seq samples were converted to gff3 files using gffread v0.12.7 (RRID:SCR_018965) [54], combined, and then sorted using gff3_sort v2.1.0 [55].

Before protein sequences were aligned to the Dakapo genome assembly, repeats in the assembly were masked using RepeatMasker v4.1.2-p1 (RRID:SCR_012954) [56] with the TE library generated using EDTA [44] and the following flags: *-e rmblast, -s, -norna, -xsmall, -gff, -html,* and *-source*. Protein sequences from the Arabidopsis (*Arabidopsis thaliana*) Araport11 annotation [57], the *Oryza sativa* Release 7 annotation [58], and the Viridiplantae UniProtKB/Swiss-Prot reviewed protein sequence dataset from UniProt release 2023_05 [59] were aligned to the masked Dakapo genome assembly using exonerate v2.4.0 (RRID:SCR_016088) [60] with the following flags: *−−model protein2genome, −−bestn 5, −−minintron 10, −−maxintron 5000, −−querychunktotal 5, −−targetchunktotal 10, −−showtargetgff yes, −−showalignment no, −−showvulgar no, −−ryo "%qi length=%ql alnlen=%qal\n>%ti length=%tl alnlen=%tal\n"*. The outputs for each dataset were combined, reformatted using the custom script reformat_exonerate_protein_gff.pl, and sorted using gff3_sort v2.1.0 [55].

MAKER v3.01.04 [61] was initially run on the Dakapo genome assembly with the gff files generated through transcript assembly and protein sequence alignment. These initial annotations generated by MAKER were then used to train SNAP (RRID:SCR_007936) and AUGUSTUS (RRID:SCR_008417). To train SNAP, maker2zff from MAKER was first used to convert genes to the ZFF format with the flag *-x 0.1*. This input was used with SNAP v2013_11_29 [62] to first categorize genes by running the command *fathom* to produce reformatted files, followed by the command *forge* to estimate parameters. Hidden Markov Models (HMMs) were created using *hmm-assembler.pl* from SNAP [62]. To train AUGUSTUS, maker2zff from MAKER was first used to convert genes to the ZFF format with the following flags: *-c 0.5, -e 0.5, -o 0.5, -a 0, -t 0, -l 200,* and *-x 0.2*. The *fathom* command from SNAP [62] followed by the custom script fathom_to_genbank.pl were then run to reformat the files and keep only 600 randomly sampled annotations. Fasta files of the subsetted genes were then generated using the custom script get_subset_of_fastas.pl. These subsetted genes were split into training and test files, and then *autoAug.pl* from AUGUSTUS v3.4.0 [63] was run to produce batch scripts that were then run. This step was repeated using the following flags with *autoAug.pl*: *-useexisting* and *−−index=1*. The sensitivity and specificity of the AUGUSTUS HMMs were evaluated by running the *augustus* command. A second round of MAKER



v3.01.04 [61] was then run using the HMMs from SNAP and AUGUSTUS to produce gene annotations.

We then filtered annotations and flagged genes that may have actually been transposons annotated as genes using methods described previously [64]. To ensure that our annotations were as complete as possible, we used Liftoff v1.6.2 [65] to transfer annotations from the PN40024.v4 grapevine genome assembly [66] to the Dakapo genome. We then used the methods described previously [64] to assign "pseudogene" and "gene" labels to the lifted genes based on the confidence of the lifted gene model.

Finally, gene functions were assigned to each annotated gene by first using InterProScan v5.66-98.0 (RRID:SCR_005829) [67] to assign Pfam domains and corresponding gene ontology (GO) terms using the following flags: *-appl pfam, -goterms, -pa, -dp, -iprlookup, -t p*, and *-f TSV*. Then, Arabidopsis orthologs were identified by running DIAMOND v2.0.15.153 (RRID:SCR_009457) [68] with protein sequences from Dakapo and the TAIR10 Arabidopsis annotation [69] and the following flags: *--evalue 1e-6, --max-hsps 1, --max-target-seqs 5*, and *--outfmt 0*. The results from InterProScan, DIAMOND, and Arabidopsis gene functions and GO terms of orthologs [70] were all combined to generate a file with functional descriptions for each gene using the custom script create_functional_annotation_file.pl.

## DNA extraction and sequencing of Rubired tissue

High-molecular-weight genomic DNA was extracted using the method previously described [71]. The PacBio highly accurate long reads (HiFi) library preparation and sequencing were performed as previously described [72]. The HiFi library fraction with a length >15 kbp was sequenced in two SMRT cells on a PacBio Sequel IIe platform at the DNA Technology Core Facility, University of California, Davis. The sequencing generated 31.3 Gbp sequences corresponding to 62.6× coverage with an N50 of 11.5 kbp.

## Rubired genome assembly

The pseudomolecules of the Rubired genome were assembled, phased, and scaffolded using methods described previously [72]. Briefly, after testing multiple Hifiasm v.0.16.1-r374 (RRID:SCR_021069) [73] parameters, the best assembly obtained with the configuration '*-a 4 -k 41 -w 71 -f 25 -r 4 -s 0.7 -D 3 -N 100 -n 25 -z 20*' was selected: it consisted of 273 contigs with an N50 = 12.9 Mb. An integrated phasing and scaffolding procedure further led to the construction of chromosome-scale pseudomolecules using HaploSync [74] combined with a high-density consensus map [75]. Two runs of HaploSplit were performed, followed by two runs of the HaploFill module [74]. The quality and completeness of the assembly for the Dakapo genome were assessed as described above.

## Rubired genome annotation

The structural and functional annotation of the Rubired genome followed the exhaustive annotation pipeline previously described [76]. Briefly, high-quality Iso-Seq data from *V. vinifera* Cabernet Sauvignon [47], quality-based filtered RNA-Seq data from *V. rupestris* [77], and external databases were used to generate a collection of assemblies, alignments, *ab initio* predictions, and transcript/protein evidence. High-quality gene models were generated using PASA v2.3.3 (RRID:SCR_014656) [78] for training gene predictors, including Augustus v.3.0.3 [79], GeneMark v.3.47 (RRID:SCR_011930) [80], and SNAP v.2006-07-28 [62]. *Ab initio* predictions were produced using the aforementioned tools and

BUSCO v.3.0.2 [81]. In parallel, repeat annotations were obtained using RepeatMasker v.open-4.0.6 [56]. Combined with the transcript alignments from PASA and protein alignments generated with Exonerate v.2.2.0 [60], all forms of evidence were merged into consensus gene models using EVidenceModeler v.1.1.1 (RRID:SCR_014659) [82]. Finally, functional annotations were attributed with Blast2GO v.4.1.9 (RRID:SCR_005828) [83], using results from DIAMOND blastp v.2.0.15.153 [68] against the Refseq plant protein database [84] and InterProScan v.5.28-67.0 [67].

## Investigating the *VvMybA1* sequences in the Dakapo and Rubired genomes

Teinturier grape varieties are known to have tandem copies of a 408 bp sequence within the promoter region of *VvMybA1*, a key gene in anthocyanin biosynthesis that leads to increased anthocyanin accumulation within their berries. Dakapo and Rubired contain three and two copies of this repeat, respectively [11]. To investigate the sequence similarities of these repeat sequences in our assembled genomes, we first used blastn (RRID:SCR_001598) from BLAST v2.10.0+ (RRID:SCR_004870) [85] to search for the locations of *VvMybA1* within the Dakapo genome and both Rubired haplotypes. We then extracted the sequences of *VvMybA1* and the 10 kbp surrounding the gene from the genomes using BEDTools v2.27.1 (RRID:SCR_006646) *getfasta* [86]. We searched these sequences for the 408 bp repeat identified previously [11] both manually and using blastn from BLAST v2.10.0+ [85].

## Exploring synteny among various grapevine genomes

GENESPACE v1.3.1 [87] was used with MCScanX (RRID:SCR_022067) [88] and Orthofinder v2.5.5 (RRID:SCR_017118) [89] to align and plot protein sequences from the following chromosome-scale grapevine genomes: Dakapo, Rubired, Cabernet Franc (FPS clone 04) [74], Cabernet Sauvignon (FPS clone 08) [74], Chardonnay (FPS clone 04) [90], and Pinot Noir (FPS clone 123) [91]. Individual chromosomes from the Dakapo and Rubired assemblies were aligned using MUMMer v4.0.0rc1 (RRID:SCR_018171) [92]. To do so, first, the *nucmer* command was run with default settings. The command *delta-filter* was then run with the following flags: *-i 90 -l 5000*. Finally, plots were generated using the *mummerplot* command.

## DATA DESCRIPTION AND QUALITY CONTROL
### The Dakapo genome assembly and annotation

The Dakapo genome was assembled using ONT reads representing 142.4× coverage (based on a genome size of 500 Mbps). This draft assembly was then scaffolded to the 12X.v2 grapevine reference genome [32] and polished using both ONT reads (142.4× coverage) and Illumina reads (51.3× coverage) to produce the final genome assembly of 508.5 Mbp. The final genome assembly comprises 19 chromosomes and 542 unplaced contigs, with 96.3% of the Dakapo assembly sequence located on the chromosomes and 2,644 gaps of unknown sequence. The final genome assembly is highly contiguous, with an N50 of 25.6 Mbp, slightly higher than the PN40024.v4 assembly [66] and similar to the most recent PN40024 telomere-to-telomere (PN_T2T) assembly [93]. The Dakapo assembly has a high BUSCO score of 97.7% complete BUSCOs (94.5% single-copy BUSCOs and 3.2% duplicated BUSCOs), similar to prior PN40024 reference assemblies (97.8% for 12X.v2 [32], 98.3% for PN40024.v4 [66], and 98.4% for PN_T2T [93]). In addition, the Dakapo genome received a raw LAI score of



**Table 1.** Assembly statistics of the Dakapo and Rubired assemblies, along with previous grapevine reference genome assemblies. Comparison of Dakapo and Rubired genome assembly and annotation statistics with previous grapevine reference genomes (12X.v2 [32], PN40024.v4 [66], and PN_T2T [93]). The Rubired whole assembly contains both haplotypes and unplaced sequences.

|  | Dakapo | Rubired whole assembly | Rubired haplotype-1 | Rubired haplotype-2 | 12X.v2 [32] | PN40024.v4 [66] | PN_T2T [93] |
|---|---|---|---|---|---|---|---|
| **Assembly size (Mbp)** | 508.5 | 983.8 | 476.0 | 474.7 | 486.2 | 475.6 | 494.9 |
| **Number of contigs** | 561 | 185 | 19 | 19 | 20 | 22 | 19 |
| **N50 (Mbp)** | 25.6 | 24.9 | 24.7 | 24.9 | 24.3 | 24.4 | 25.9 |
| **Number of gaps** | 2,644 | 97 | 38 | 59 | 15,325 | 4,019 | 0 |
| **Total complete BUSCO** | 97.7% | 98.7% | 98.3% | 97.3% | 97.8% | 98.3% | 98.4% |
| **Raw LAI** | 12.22 | N/A* | 15.22 | 15.62 | 9.4** | 13.97 | 14.29 |
| **Genes annotated** | 36,940 | 56,681 | 27,586 | 27,799 | 42,414 | 35,230 | 37,534 |

*The raw LAI score was not calculated for the whole assembly due to high sequence similarity between haplotypes, which would prevent an accurate calculation. **Previously calculated [94].

12.22 and thus contains a reference-quality assembly of repetitive/intergenic sequences [94] (Table 1).

The Dakapo genome was annotated using a combination of *de novo* annotations using MAKER [61] and annotations lifted from PN40024.v4 [66] using Liftoff [65]. This resulted in 36,940 genes being annotated. We also annotated both TEs and repeat sequences and found that these comprised 45.38% of the genome, similar to what was previously reported in grapevine (41.4–51.1% [90, 95, 96]). LTRs make up a majority of the repetitive sequences annotated in the Dakapo genome and comprise 30.48% of the genome, with *Gypsy* LTRs specifically being the most abundant type, comprising 12.88% of the Dakapo genome sequence (Supplementary Table S3 on GigaDB [25]).

## The Rubired genome assembly and annotation

The Rubired genome was sequenced with highly accurate long-read sequencing, generating 62.6× HiFi coverage (using a haploid genome size of 500 Mbp as reference). Pseudomolecules were constructed by scaffolding and phasing the assembly using HaploSync [74], generating two haplotypes comprising 19 chromosomes and averaging a total length of ~475 Mbp. With complete BUSCO scores of 98.3% and 97.3% for haplotype-1 and haplotype-2, respectively (between 95.5–96.3% single copy BUSCOs and between 1.8–2.0% duplicated BUSCOs, respectively), and only 33 Mbp of unplaced sequences in the diploid assembly of the Rubired genome, the Rubired assembly is highly complete. Both genomes for the two Rubired haplotypes also have high raw LAI scores (15.22 for haplotype-1 and 15.62 for haplotype-2), demonstrating that the diploid Rubired genome contains a reference-quality assembly of repetitive/intergenic sequences that are likely more complete than the 12X.v2 [32], PN40024.v4 [66], and PN_T2T [93] assemblies (Table 1). The gene annotation resulted in 56,681 genes for the whole assembly, showing a chromosome anchoring of 97.7%, further supporting the reference quality of the assembly. A similar number of genes was annotated for both haplotypes (27,586 for haplotype-1 and 27,799 for haplotype-2), and very few were annotated on unplaced contigs (1,296). Overall, the genome was composed of 50.46% repetitive sequences with a clear accumulation in the unplaced sequences, with 74.34% of its sequences annotated as repeats. The repeat distribution was similar to the Dakapo genome, with *Gypsy* LTRs as the predominant repeats type, corresponding to 13.91% of the genome sequence (Supplementary Table S4 on GigaDB [25]).

## RE-USE POTENTIAL

Grapevine varieties have been bred to produce berries in a variety of colors, commonly divided into red-, black-, and white-skinned berries that typically have white flesh. However, there are several varieties of teinturier grapes, which contain pigmented skin and pigmented flesh, including the Dakapo and Rubired varieties sequenced here. Here, we present high-quality genome assemblies and annotations for these two teinturier grape varieties. These two genomes were generated using distinct sequencing technologies, resulting in different assembly/annotation methods being used. As a result, the Rubired reference genome is haplotype-resolved and both more contiguous and complete than the Dakapo reference genome. Nonetheless, both assemblies are highly contiguous and complete, and will greatly facilitate future research. By assembling these genomes, we fully assembled *VvMybA1* and the tandem repeat associated with anthocyanin content in teinturier grapes [10]. As expected, we found three tandem copies of this repeat within the promoter region of *VvMybA1* (the *VvMybA1t3* allele) in the Dakapo genome, exactly as described previously [10]. All three repeats contain identical 408 bp repeat sequences (Figure 2A). In addition, the Rubired haplotype-2 assembly contained two tandem copies of this repeat at the exact expected location (the *VvMybA1t2* allele) [10], with both copies containing the same 408 bp sequence as those in Dakapo (Figure 2B). The Rubired haplotype-1 assembly did not contain teinturier-associated alleles but instead contained the *VvMybA1a* allele responsible for white berry skin color [13], as expected based on previous findings [10]. The *VvMybA1a* allele is distinct from teinturier alleles and other functional *VvMybA1* alleles due to the presence of the *Gret1* retrotransposon upstream of coding sequences [13]. However, it does contain the 408 bp repeat upstream of the *Gret1* retrotransposon [10]. This repeat sequence is not perfectly identical to the repeat in Dakapo or Rubired haplotype-2 and instead contains three single base pair mutations within the sequence (Figure 2C).

Beyond fully sequencing the *VvMybA1* alleles of Dakapo and Rubired, these genomes will enable more insight into grapevine berry color by providing two high-quality teinturier grapevine genomes for future studies. As previously mentioned, teinturier grapes differ in the composition of total anthocyanins produced, and this phenomenon does not seem to be driven by differences in *VvMybA1* alleles [12]. These genomes will provide resources for investigating the genetic mechanisms driving this phenomenon. Focusing on the berry color locus on chromosome 2 [97–99] and the anthocyanin locus on chromosome 14 [100], in particular, may provide insight into the regulation of specific anthocyanin molecules within the flesh of teinturier berries.

The Dakapo and Rubired genomes and annotations will also offer additional resources for future work in grapevines. Beyond berry flesh color, the Dakapo and Rubired genomes will also provide resources for investigating additional traits. For example, Dakapo is both frost-susceptible [101] and *Botrytis*-susceptible [19], while Rubired is notably highly mildew-resistant [102]. A QTL-mapping population generated through a cross between Dakapo × Cabernet Sauvignon has also been previously established [19], so the availability of this reference genome will greatly aid future studies with this population. These genomes will ultimately provide new resources for investigating a variety of grapevine traits, enabling advances in grapevine breeding and agriculture and allowing for comparisons between grapevine genomes (Figure 3).

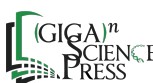

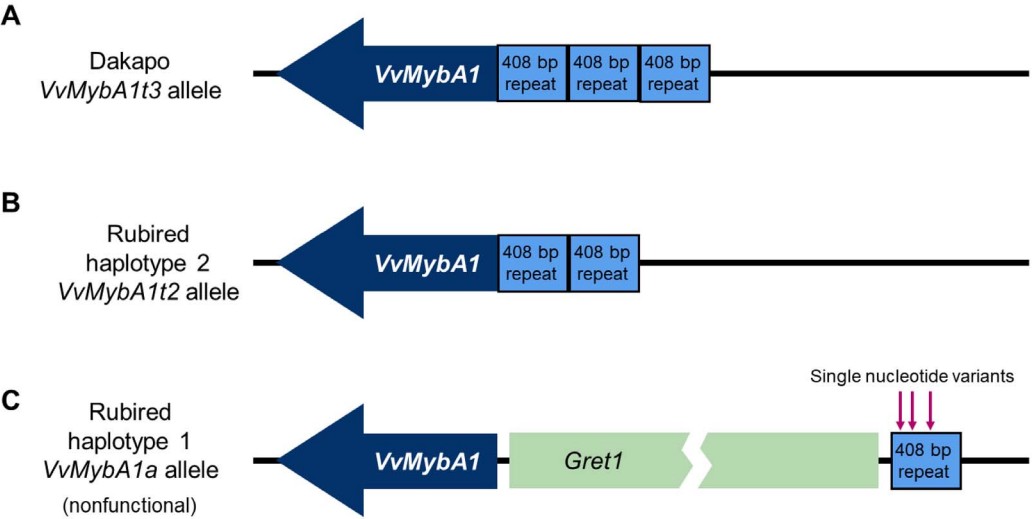

**Figure 2.** Diagrams of the *VvMybA1* alleles in (A) the Dakapo assembly, (B) the Rubired haplotype-2 assembly, and (C) the Rubired haplotype-1 assembly. *VvMybA1* is represented by the dark blue arrow, and the 408 bp repeats are shown in light blue boxes. Dakapo contains three tandem copies of the 408 bp repeat, while the Rubired haplotype-2 assembly contains two tandem copies. The Rubired haplotype-1 assembly contains the nonfunctional *VvMybA1a* allele with the *Gret1* retrotransposon shown upstream of *VvMybA1* in a light green box, truncated to fit in the figure. The three single nucleotide variants within the 408 bp repeat of the nonfunctional allele in Rubired haplotype-1 are indicated by arrows.

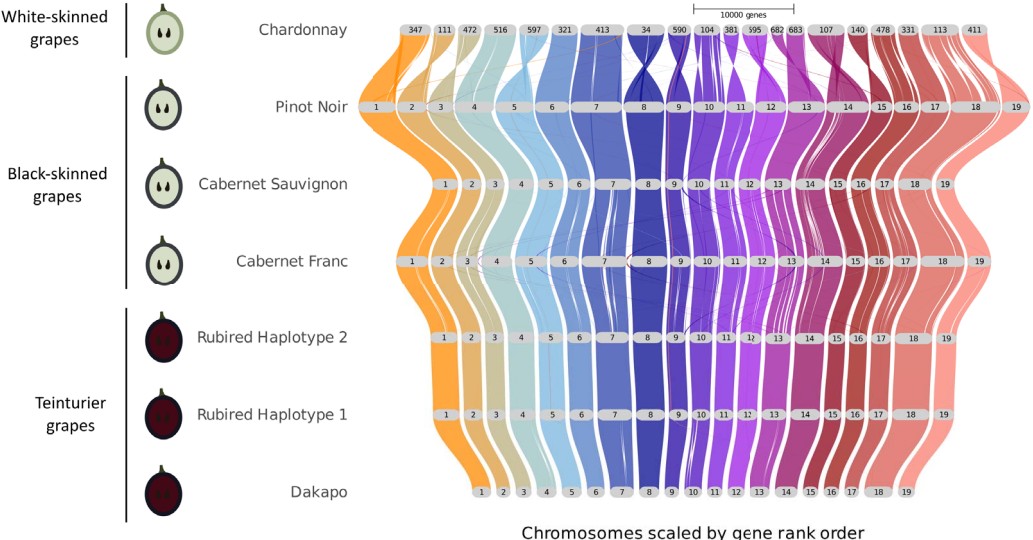

**Figure 3.** Synteny of several grapevine genomes with chromosome-scale assemblies, organized by berry and flesh color.

Initial comparisons of these assemblies to other grapevine genomes even revealed a putative large (1.82 Mbp) inversion on chromosome 10 within Dakapo (Figure 4) that contains 274 genes (Supplementary Table S5 on GigaDB [25]). This inversion appears to be absent in the other chromosome-scale grapevine-genome-assemblies compared in Figure 3. To ensure that this putative inversion in Dakapo was not due to scaffolding errors, we split



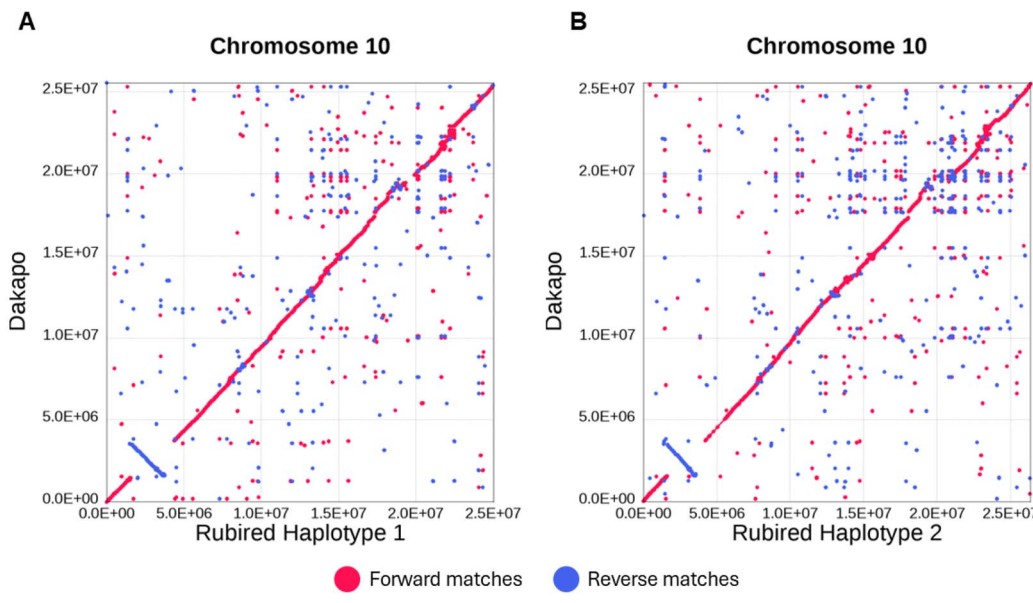

**Figure 4.** Alignment of chromosome 10 in Dakapo versus chromosome 10 in (A) the Rubired haplotype-1 assembly and (B) the Rubired haplotype-2 assembly, showing the putative 1.82 Mbp inversion present in Dakapo. The dotplots show forward matches in red and reverse matches in blue.

apart chromosome 10 of Dakapo at gaps (introduced through scaffolding) to verify that no gaps were near the predicted inversion breakpoint and that these contigs still showed evidence of being inverted compared to other genomes. No assembly gaps were near the inversion breakpoints, with the closest gaps being ~600 kbp and ~1.3 Mbp away from the inversion breakpoints. Aligning contigs from the split apart chromosome 10 of Dakapo to chromosome 10 of the PN_T2T reference [93], using MUMmer v4.0.0rc1 [92] to produce dotplots as previously described, also showed evidence for linear yet inverted alignments in the region of the putative inversion (Supplementary Figure 1 on GigaDB [25]). To further verify the presence of the inversion, we mapped the trimmed Dakapo Illumina paired-end sequencing data used to assemble the genome to the Dakapo reference genome using bwa-mem2 v2.2.1 (RRID:SCR_022192) [103], marked duplicates using Picard v2.15.0 [36], and filtered the mapped reads using SAMtools v1.17 (RRID:SCR_002105) [104]. These paired-end reads mapped as expected to the Dakapo reference genome, with nearly all reads having proper insertion sizes and orientations, supporting the presence of this putative inversion (Supplementary Figures 2 and 3 on GigaDB [25]). Ultimately, the breakpoints of the putative inversion will need to be validated using PCR amplification and Sanger sequencing. This would also confirm if the putative inversion is heterozygous or homozygous in Dakapo.

Inversions can cause changes in gene expression depending on various genetic factors [64, 105, 106]; hence, we were interested in whether the inversion of the putative Dakapo chromosome 10 could contribute to Dakapo's increased cold susceptibility and/or increased pathogen susceptibility. Several genes within the inversion do appear to be involved in cold- and/or pathogen-responsive pathways, including VvDak_v1.10g0003381, whose *Arabidopsis* ortholog (AT3G07650) regulates the expression of genes within the cold acclimation pathway [107], and VvDak_v1.10g0003951, whose *Arabidopsis* and rice orthologs (AT4G03960 and *OsPFA-DSP2*, respectively) negatively regulate pathogen

response pathways [108]. The implications of this inversion remain unclear; however, future research could unveil the potential phenotypic impacts of this putative inversion.

Grapevine is a useful model system due to its unique life and domestication history and is one of few lianas (woody vines) with robust genomic resources. In addition, grapevine breeding and propagation have been ongoing for millennia, resulting in a fascinating array of phenotypes and an abundance of accumulated somatic variants. The assemblies and annotations of the Dakapo and Rubired genomes add to a growing number of grapevine genomes that will provide valuable tools for both grapevine breeders and geneticists.

## AVAILABILITY OF SOURCE CODE AND REQUIREMENTS
- **Project name:** Genome assembly and annotation for two teinturier grapevine varieties
- **Project home page:** https://github.com/eleanore-ritter/teinturier-grapevine-genomes
- **Operating system:** Platform-independent
- **Programming languages:** bash, perl, python, and R
- **Other requirements:** packages described in methods
- **License:** Apache-2.0.

## DATA AVAILABILITY
The genomes and annotations for both Dakapo and Rubired are available on grapegenomics.com, Zenodo [108, 109], and GigaDB (which also includes snapshots of the code) [25]. Sequencing data from this study are provided on the NCBI Sequence Read Archive under BioProjects PRJNA1094988 and PRJNA1085245. Supplementary tables and figures are available on GigaDB [25].

## LIST OF ABBREVIATIONS
EDTA, Extensive de-novo transposable element Annotator; FPS, Foundation Plant Services; GCE, grapevine color enhancer; GO, gene ontology; HiFi, highly accurate long reads; HMMs, Hidden Markov Models; LAI, long terminal repeats Assembly Index; LTR, long terminal repeats; ONT, Oxford Nanopore Technologies; PN_T2T, PN40024 telomere-to-telomere; SRA, Sequence Read Archive; TE, transposable element.

## DECLARATIONS
### Ethics approval and consent to participate
Not applicable.

### Consent for publication
Not applicable.

### Competing interests
Peter Cousins is an employee of E. & J. Gallo Winery.

### Authors' contributions
CN envisioned the project, secured the funding, and supervised research on Dakapo. DC envisioned the project, secured the funding, and supervised research on Rubired. EJR assembled and annotated the Dakapo genome, designed and executed comparative analyses within the study, and wrote the first draft of the manuscript. NC and AM

assembled and annotated the Rubired genome. PC led Dakapo vine cultivation and tissue sample collection and helped conceptualize comparative analyses within the study. All authors assisted with the final draft of the manuscript.

## Funding

The Dakapo genome work was supported by Michigan State University and the USDA National Institute of Food and Agriculture MICL02572. Oxford Nanopore Technologies provided sequencing for the Dakapo assembly. The Rubired genome was supported by E. & J. Gallo Winery and the NSF grant #1741627.

## Acknowledgements

We are grateful to Dan Chitwood, Emily Josephs, and Robin Buell for helpful discussions on this work and feedback on this manuscript. We are grateful to the Genomics Core at Michigan State University, the Institute for Cyber-Enabled Research at Michigan State University, and the UC Davis DNA Technology Core for their services. We would like to thank Kevin Childs for providing guidance and custom scripts for genome annotation. We acknowledge Rosa Figueroa-Balderas for processing the samples, extracting the nucleic acids, and preparing the sequencing libraries for the Rubired genome. Lastly, we are grateful to the reviewers for their thorough review and suggestions, which have greatly strengthened the manuscript.

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
