## [Editor Report]

Editor’s AssessmentTeinturier grapes produce berries with pigmented skin and flesh, and are used in red wine blends, as they provide a deeper colour. This paper presents the genomes of two popular teinturier varieties (Dakapo and Rubired); sequenced, assembled, and annotated to provide additional resources for their use in breeding. Combining Nanopore and Illumina sequencing for Dakapo, scaffolding to the existing grapevine assembly to generate a final assembly of 508.5 Mbp and 36,940 gene annotations. For Rubired PacBio HiFi reads were assembled, scaffolded, and phased to generate a diploid assembly with two haplotypes 474.7-476.0 Mbp long and 56,681 genes annotated. Peer review has helped validate their high quality, these genomes hopefully enabling more insight into the genetics of grapevine berry colour and their other traits like frost and mildew-resistance.Editor’s AssessmentTeinturier grapes produce berries with pigmented skin and flesh, and are used in red wine blends, as they provide a deeper colour. This paper presents the genomes of two popular teinturier varieties (Dakapo and Rubired); sequenced, assembled, and annotated to provide additional resources for their use in breeding. Combining Nanopore and Illumina sequencing for Dakapo, scaffolding to the existing grapevine assembly to generate a final assembly of 508.5 Mbp and 36,940 gene annotations. For Rubired PacBio HiFi reads were assembled, scaffolded, and phased to generate a diploid assembly with two haplotypes 474.7-476.0 Mbp long and 56,681 genes annotated. Peer review has helped validate their high quality, these genomes hopefully enabling more insight into the genetics of grapevine berry colour and their other traits like frost and mildew-resistance.

---

## [Reviewer Report]

Indicate in the comments box below whether you are happy with the changes made or if the manuscript is unacceptable.Comments on revised manuscriptIn Line 53, There are differing conclusions regarding the history of grapes, and certain disagreements persist, such as doi: 10.1073/pnas.2222041120, doi: 10.1073/pnas.1709257114. Therefore, it is essential to provide an objective summary and comprehensive citations. Additionally, the statement that grapes are primarily used for winemaking is inaccurate, as they are also widely consumed fresh.

---

## [Reviewer Report]

Reviewer name and names of any other individual's who aided in reviewer Camille RustenholzDo you understand and agree to our policy of having open and named reviews, and having your review included with the published papers. (If no, please inform the editor that you cannot review this manuscript.)YesIs the language of sufficient quality?YesPlease add additional comments on language quality to clarify if needed
Are all data available and do they match the descriptions in the paper? YesAdditional CommentsAre the data and metadata consistent with relevant minimum information or reporting standards? See GigaDB checklists for examples <a href="http://gigadb.org/site/guide" target="_blank">http://gigadb.org/site/guide</a>YesAdditional CommentsIs the data acquisition clear, complete and methodologically sound?YesAdditional CommentsIs there sufficient detail in the methods and data-processing steps to allow reproduction?NoAdditional CommentsOverall, the authors give enough details except for the haplotypes of Chardonnay, Pinot noir, Cabernet sauvignon and Cabernet franc that were used for Figure 3.Is there sufficient data validation and statistical analyses of data quality? YesAdditional CommentsIs the validation suitable for this type of data?NoAdditional CommentsOverall, the authors provide accurate validation for this type of data except for the inversion that was identified on chromosome 10 of Dakapo assembly. In my opinion, more evidences need to be provided as Dakapo contigs were anchored using PN40024 12X.v2 assembly version. There is indeed a heterozygous region at the beginning of chromosome 10 in PN40024 genome which makes its assembly and scaffolding quality quite doubtful at that exact location and especially for this assembly version. I would suggest to check it using the latest PN40024 T2T version (Shi et al., Hort Res 2023) and to show some Dakapo short read alignments against its own assembly to validate the borders of this inversion, even though some wet lab validation would be even more convincing.Is there sufficient information for others to reuse this dataset or integrate it with other data?YesAdditional CommentsAny Additional Overall Comments to the AuthorThe authors provided the assemblies and gene annotations of the genomes of two teinturier varieties, Dakapo and Rubired. Dakapo was assembled using a combination of Nanopore and Illumina reads whereas Rubired was assembled using PacBio HiFi reads. Even though both assemblies are of high quality, quality metrics are better for Rubired assembly than for Dakapo assembly, in terms of contiguity and of phasing. I would have liked the authors to comment and explain these differences more extensively maybe in a dedicated paragraph in the Discussion section: - Why Dakapo assembly could not be phased? - Are these differences in terms of quality due to the sequencing technologies (Nanopore versus PacBio HiFi) used? Or to different year of dataset acquisition? Or to assembly methods? Both assemblies were also annotated: 36,940 genes in the Dakapo assembly and 56,681 genes in the diploid Rubired. I assume that 56,681 is the sum of the number of genes annotated on haplotype 1 and haplotype 2 of Rubired. If so, it needs to be clearly stated line 328 otherwise it can be confusing for the reader who will think that Rubired has 20,000 more genes than Dakapo. Also, the authors used two different annotation pipelines, which complicates the gene content comparison and the synteny analysis later on. I would have liked the authors to comment and explain it: - Is it due to the difference in the quality of the assemblies? If so, the authors need to highlight the limits of their annotation pipeline regarding assembly quality. - Any other explanation? Some minor suggestions : - Line 74: please use the word “clone” in the sentence for a matter of clarity. - Line 292-293: PN40024.v4 assembly is not the most recent but the PN40024 T2T is (Shi et al., Hort Res, 2023) The quality of the assemblies and annotations are very good and the resources of the paper will be very valuable for the grapevine community, especially to study the anthocyanin production in grapevine.RecommendationMinor Revision

---

## [Reviewer Report]

Upload additional filesDRR-202406-01-R02/stage_files/DRR-202406-01/Review MS/Ritter_et_al._2024_Gigabyte_reviewer_comments_8-23-24.docxReviewer name and names of any other individual's who aided in reviewer Andrea GschwendDo you understand and agree to our policy of having open and named reviews, and having your review included with the published papers. (If no, please inform the editor that you cannot review this manuscript.)YesIs the language of sufficient quality?YesPlease add additional comments on language quality to clarify if needed
Are all data available and do they match the descriptions in the paper? NoAdditional CommentsThe supplementary files were not made available to me for review.Are the data and metadata consistent with relevant minimum information or reporting standards? See GigaDB checklists for examples <a href="http://gigadb.org/site/guide" target="_blank">http://gigadb.org/site/guide</a>YesAdditional CommentsIs the data acquisition clear, complete and methodologically sound?YesAdditional CommentsIs there sufficient detail in the methods and data-processing steps to allow reproduction?YesAdditional CommentsI recommend including additional details for the programs used for the Rubired genome assembly and annotation in this manuscript, though.Is there sufficient data validation and statistical analyses of data quality? NoAdditional CommentsIt is unclear from the manuscript if the large Dakapo inversion was validated experimentally.Is the validation suitable for this type of data?YesAdditional CommentsIs there sufficient information for others to reuse this dataset or integrate it with other data?YesAdditional CommentsAny Additional Overall Comments to the AuthorPlease see attached Word documentRecommendationMajor Revision

---

## [Reviewer Report]

Reviewer name and names of any other individual's who aided in reviewer Kekun ZhangDo you understand and agree to our policy of having open and named reviews, and having your review included with the published papers. (If no, please inform the editor that you cannot review this manuscript.)YesIs the language of sufficient quality?YesPlease add additional comments on language quality to clarify if needed
Are all data available and do they match the descriptions in the paper? NoAdditional CommentsAre the data and metadata consistent with relevant minimum information or reporting standards? See GigaDB checklists for examples <a href="http://gigadb.org/site/guide" target="_blank">http://gigadb.org/site/guide</a>YesAdditional CommentsIs the data acquisition clear, complete and methodologically sound?YesAdditional CommentsIs there sufficient detail in the methods and data-processing steps to allow reproduction?YesAdditional CommentsIs there sufficient data validation and statistical analyses of data quality? NoAdditional CommentsIs the validation suitable for this type of data?YesAdditional CommentsIs there sufficient information for others to reuse this dataset or integrate it with other data?NoAdditional CommentsAny Additional Overall Comments to the AuthorMy main concerns: 1. Please explain why different sequencing methods were chosen for the genome assembly of Dakapo and Rubired, given that HiFi sequencing is currently mainstream and provides more accurate assembly? 2. Recently, the T2T level genome of many grape cultivars has been assembled including the reference genome PN_T2T and the teinturier grape Yan73, Please align with the latest complete reference genome PN_T2T in Line 172, and add the genome information about PN_T2T and Yan73 in Table 1. ( DOI10.1093/hr/uhad061, DOI10.1093/hr/uhad205 ) 3. Line 387-389: How did you verify the correctness of this inversion? Is it contained within a single contig without orientation or assembly errors in the Dakapo genome? Have you identified any other genomes with this inversion? 4. Line 255: can you explain why is the contig N50 so low? 5. Line 328: whether the total number of annotated genes in the two Rubired haplotypes are all 56,681? it would be more appropriate to describe them separately. 6. The phenotypes of these two grapes should be included, not just in the pattern diagram. 7. The sequence difference in Figure 2 should be verified using other methods, such as PCR results and Sanger sequencing.RecommendationMajor Revision